# Intersectional Gaps in Self-Efficacy among Post-Graduate Students in International Renewable-Energy Programs: The Role of Maternal Employment

**Marcella Corsi** [1], **Giulia Zacchia** [1,*] and **Izaskun Zuazu** [2]

1   Department of Statistics, Sapienza University of Rome, 00161 Rome, Italy; marcella.corsi@uniroma1.it
2   Institute for Socio-Economics, University of Duisburg-Essen, 47057 Duisburg, Germany; izaskun.zuazu-bermejo@uni-due.de
*   Correspondence: giulia.zacchia@uniroma1.it

**Abstract:** Relatively little empirical research has analyzed the sources of students' self-perceptions outside the US and Europe, and in new fields of study like renewable energy. This paper aims at filling this gap by identifying differences in self-efficacy levels of post-graduate students in Erasmus+ capacity-building programs on renewable energy in Argentinian and Guatemalan universities. We analyzed a sample of 43 students to test intersectional differences in self-efficacy, looking at students' gender, country of origin, and maternal employment. Using the New General Self-Efficacy scale, we performed the *t*-test to compare mean differences in self-efficacy, and one-way and two-way ANOVA tests to check the consistency of the results. Our estimates did not show significant gender gaps in self-efficacy among renewable-energy post-graduate students, but they did uncover relevant country differences in mean self-efficacy levels, mainly due to differences in socio-economic indicators and gender norms between the two countries analyzed. Moreover, we found a mediating role of maternal employment in cross-country self-efficacy differences, whereas the characteristics of fathers appeared uninfluential. We conclude by stressing the importance of intersectional analysis in terms of country of origin, family backgrounds, and gender norms to increase knowledge about differences in self-efficacy of students.

**Keywords:** self-efficacy in higher education; gender gaps; maternal employment; Argentina; Guatemala



## 1. Introduction

The role of students' self-efficacy, or personal beliefs about their own capabilities, in the learning process has long been investigated (van Dinther et al. 2011; van den Heuvel et al. 2015). Extensive empirical evidence suggests a strong link between self-efficacy and student achievement (van Dinther et al. 2011; Siriparp 2015; Bartimote-Aufflick et al. 2016; Johnson and Muse 2017). Similarly, self-efficacy is proven to foster professional career and job prospects of individuals (Stajkovic and Luthans 1998), and it provides motivational benefits in job satisfaction and turnover intentions (Ozyilmaz et al. 2018).

Although self-perception is usually related to student performance both within higher education systems and at workplaces, relatively little empirical research has analyzed the sources of students' self-perceptions outside the US and Europe. Researchers have found that teaching practices and classroom climate (Carol et al. 2001), as well as students' personal characteristics such as gender (Pajares and Miller 1994; Pajares 2002), ethnicity (Cuellar 2014) and parents' education (Santiago and Einarson 1998; Riggio and Desrochers 2006), are associated with higher student self-efficacy.

This paper extends those efforts by widening the breadth of the research focus to contexts different from the US and Europe, as recommended in Roy et al. (2019), and analyzing a new field of study: renewable energy (RE). The study, in fact, aims to identify sources of differences in self-efficacy levels in the context of higher education systems. To do so, we

collected data about a group of participants in an Erasmus+ training program on renewable energy called DIEGO (Development of quality system through energy efficiency courses; for more info about the project, visit the web site: http://www.diego-energy.eu/results, Accessed on: 1 June 2021). The program was part of a capacity-building project in the field of higher education in engineering, financed by the Education, Audiovisual, and Culture Executive Agency (EACEA) of the European Commission; its goal is to encourage cooperation between the EU and Latin America by developing new and innovative education programs that support the modernization, accessibility, and internationalization of higher education. The main goal of DIEGO is the implementation of new and innovative post-graduate courses in universities located in Argentina and Guatemala (the universities involved in our study were Universidad Nacional de San Luis—Argentina, Universidad Nacional del Sur—Argentina, Universidad Nacional de Chilecito—Argentina, Universidad Rafael Landivar—Guatemala, and Universidad Galileo—Guatemala) to train future experts in the renewable and green-energy sectors. In fact, green jobs use more intensively high-level cognitive and interpersonal skills compared to non-green jobs (Consoli et al. 2016), so specific trainings are requested to prepare engineering students for these new roles in the local labor markets (van den Heuvel et al. 2015; van der Horst and Klehe 2019).

Given the widespread concerns about women's presence in graduate programs in engineering (Adelman 1998; Drew 1996; Seymour and Hewitt 1997), the first step of our study was to test specifically the existence of gender gaps in self-perception. Then we concentrated on differences among countries (Argentina and Guatemala). Finally, we studied how students' family backgrounds, in particular the presence of working mothers in formal labor markets, can impact students' self-efficacy. To identify key aspects of students involved in the DIEGO program in Argentina and Guatemala, we designed a survey (the full survey is available upon request) that focused specifically on the three main sources of differences in self-efficacy described above: gender, country of origin, and maternal employment status.

## 2. Background and Literature

Along the lines of Bandura's social cognitive theory (Bandura 1986, 1997; Bandura et al. 1996), many empirical studies drawing from learning, cognitive, and social cognitive theories were able to shed light on the nature, sources, and psychological processes involved in the formation of self-efficacy beliefs, mainly among young students (Usher and Pajares 2008).

Cognitive theorists emphasize that environmental contexts and social identities play a crucial role in creating motivational and self-regulatory differences among young people. In particular, Pajares (2002) suggested that gender differences in self-efficacy related to specific skills and academic areas deal with gender stereotypical beliefs. For example, gender gaps can arise because of social, cultural, educational, and mass-media influences that used to portray mathematics and science as masculine domains and writing or poetry within a feminine domain. This trend has been evident since the early stages of the educational path of girls and boys (Louis and Mistele 2012). Indeed, it is commonly suggested that one major consequence of differences in self-conceptions of women and men is the so-called horizontal gender segregation in higher education (Charles and Bradley 2009; Charles 2011; Ceci and Williams 2015). Self-selection of men in male-dominated fields, and of women in female-dominated ones, goes thus in line with pre-conceptions of gender gaps in cognitive abilities (see Shapiro and Wilk 1965; Arsham and Lovric 2011 for further methodological details). Huang (2013) provided a meta-analysis of research on gender differences in self-assessment of abilities and found that women displayed higher language and arts self-efficacy, while men exhibited higher mathematics, computer, and social science self-efficacy, although these differences varied with age. Consequently, as pointed out by Charles and Bradley (2009), the few women in male-dominated fields might regard themselves as "exceptional women" and as "pioneers", and the tendency might be stronger

in the educational systems, where women are overall under-represented, such as the field of engineering, which we further analyze in this paper.

In the field of engineering, Marra et al. (2009) showed, using longitudinal data, that higher self-efficacy was associated with women students' plans to persist in engineering. Importantly, their findings also suggested a relationship between ethnicity and feelings of inclusion; thus, intersectional approaches to self-efficacy might provide important insights in cognitive social theory. Litzler et al. (2014) provided further leverage of this intersectional approach. Using data on self-confidence in STEM fields, they found that the impact of personal, environmental, and behavioral factors on confidence levels hinged on demographics such as race, ethnicity, and gender.

Therefore, the social context in which students are embedded is a key element in defining drivers of imbalances in self-efficacy of students. Cultural and historical background, social environment, and ecological awareness can influence cognitions and social determinants of behavior (Luszczynska et al. 2005), and thus lead to cross-country disparities in levels of self-efficacy. Salanova et al. (2010), for example, showed that Spanish students reported higher levels of self-efficacy and engagement than did Belgian students. Morony et al. (2013) compared Confucian Asian countries with European countries and found that, on average, students in the former showed lower levels of self-efficacy than students in the latter. In a similar vein, Ahn et al. (2016) provided empirical evidence that socially conveyed sources of information relevant for self-efficacy differ across cultures.

Existing evidence also indicates that maternal employment is a fundamental socioeconomic aspect of the development of self-efficacy of students. In fact, maternal employment leads to many positive outcomes for offspring, including increased family income, greater academic achievement, fewer behavior problems, and greater positivity of the home environment (Harvey 1999). Riggio and Desrochers (2006) used a sample of 721 college students in the US and found that young adults raised by working mothers report greater positive belief in their overall competence. Nonetheless, these effects might depend on other demographics, such as race and gender (Buchanan and Selmon 2008). Consequently, an intersectional view or a cross-cultural comparison of self-efficacy levels might enable us to advance our understanding of the drivers of such a construct.

We sought to contribute to the existing literature in this direction by investigating intersectional gaps in levels of self-efficacy among students of international higher education programs in a new field of study in engineering, renewable and green energy, in Argentina and Guatemala. We explicitly considered gendered implications of international programs in RE, where women are generally a minority (Ceci and Williams 2015).

## 3. Context and Hypotheses

In contextualizing higher education in Argentina and Guatemala, we used the available comparable data, focusing on gender differences. According to United Nations Human Development Program (UNDP) data, Argentina showed a higher level of tertiary educational attainment, since the gross enrollment ratio in tertiary education was 89%, while in Guatemala it was just 22% (see Table 1). Consequently, the share of youth not in education or employment (NEETs) was lower in Argentina than in Guatemala. Instead, the phenomenon of gender horizontal segregation in different fields of study in higher education was evident in both countries: the share of female tertiary graduates in science, technology, engineering, and mathematics programs among all female tertiary graduates was 11.5% in Argentina and just 5.4% in Guatemala. Data were not available about the educational attainment in renewable and green-energy engineering courses, and actual experience involving women in RE activities in Argentina and Guatemala has been fairly limited and anecdotal to date. Documentation is sparse, and more information is needed. Still, given the opportunity, women have, in a number of cases, demonstrated their interest by taking active roles in RE projects and training activities, considering that the use of RE is quite high in Guatemala (63.7% of total final energy consumption).

**Table 1.** Argentina and Guatemala: a comparison.

| Main Educational and Occupational Data | Argentina | Guatemala |
|---|---|---|
| Higher education—Gross enrollment ratio, tertiary (% of tertiary school-age population) | 89 | 22 |
| NEETs—youth which are Neither in Employment nor in Education or Training in the youth population (% aged 15–24) | 19.3 | 27.3 |
| Female share of graduates in science, technology, engineering, and mathematics programs at the tertiary level (%) | 11.5 | 5.4 |
| Labor market | | |
| Skilled labor force (% of labor force) | 65.8 | 18.1 |
| Labor force participation rate (% aged 15 and older), male | 72.8 | 85 |
| Labor force participation rate (% aged 15 and older), female | 49 | 41.1 |
| Total unemployment rate (female-to-male ratio) | 1.27 | 1.68 |
| Youth unemployment rate (female-to-male ratio) | 1.34 | 1.82 |
| Country macro indicators | | |
| Gender Development Index (GDI) | 0.988 | 0.943 |
| Gender Inequality Index (GII) | 0.354 | 0.492 |
| Estimated gross national income per capita, female (2011 PPP USD *) | 12,085 | 4864 |
| Estimated gross national income per capita, male (2011 PPP USD *) | 23,418 | 9970 |
| Renewable energy | | |
| Renewable energy consumption (% of total final energy consumption) | 10 | 63.7 |

* PPP USD, Purchasing power parities (PPP). The conversion is used to equalise the purchasing power of the two different currencies, by eliminating the differences in price levels between the two countries. This indicator is measured in terms of national currency per US dollar. Sources: United Nations Human Development Program (UNDP). Data for each variable refer to the latest available year.

We could observe that the main differences between the countries in higher education were reflected in the labor market indicators, since Argentina had a much higher proportion of skilled labor (65.5%) than did Guatemala (5.4%). In addition, the participation of women in the labor market varied greatly between Argentina and Guatemala: in 2018, the female labor force participation rate in Argentina was 48.81%, whereas in Guatemala it was 41.06%, and the female unemployment rate compared to that of men, even for younger workers, was higher in Guatemala than in Argentina. In general, Argentina and Guatemala showed different levels of economic development as measured by gross national income per capita (in Argentina, 17,752 in 2011, expressed Purchasing power parities (PPP), compared to 7417 in Guatemala), but when looking at gender gaps in income per capita, the differences narrow, since on average in both countries women earned about 50% of what men earned.

The UNDP provides an index of gender inequality (GII) that scores countries based on three important aspects of human development (reproductive health, empowerment, and economic activity): Argentina had a GII value of 0.354, ranking it 77th out of 162 countries in the 2018 index, while Guatemala had a GII value of 0.492, ranking it 118th. This aggregated index also showed that, compared to Argentina, Guatemala had higher losses in human development due to inequality between females and males. Unfortunately, data were not available for the gender social norms index for Guatemala (the social norms index proposed in the 2019 Human Development Report comprises four dimensions—political, educational, economic, and physical integrity—and was constructed based on responses to seven questions from the World Values Survey), so it was impossible to define differences in the level of persistent discriminatory social norms between men and women in the two countries on the basis of gender inequality.

Given the main differences between Argentina and Guatemala in higher education systems and gender equality indexes, in this paper, we propose an empirical analysis to better understand the link between levels of self-efficacy and personal and social environmental backgrounds of students involved in international higher education programs in RE. Specif-

ically, for this paper we formulated the following four hypotheses regarding differences in self-efficacy scores of students in higher education in Argentina and Guatemala:

**Hypothesis 1 (H1).** *There are small or null gender differences in self-efficacy in international higher education programs in the engineering of renewable energy sector, a male-dominated field of study. Following the argument in* Charles and Bradley *(2009), women in our sample (participants in the DIEGO program) might regard themselves as "pioneers" in a gender-atypical field of study, which may provide them with a confidence in their own cognitive capabilities, and they may possess equal or higher self-efficacy than men in our sample.*

**Hypothesis 2 (H2).** *Gaps in self-efficacy follow a country's socio-economic differences. Although Guatemala and Argentina are both located in Latin America and share common cultural traits, they differ in educational, gender-related, and economic features. Following the results in* Morony et al. *(2013), we could expect to find significant differences in self-efficacy among individuals according to the country of origin.*

**Hypothesis 3 (H3).** *Individuals with employed mothers have higher levels of self-efficacy than those with homemaker mothers. In the face of the relatively scarce public support for higher education in the region (*Ferreyra et al. 2017*), we expect that participants with a dual-earner familial background might have higher levels of self-efficacy, due to higher family income levels and better access to education.)*

**Hypothesis 4 (H4).** *Country differences in self-efficacy levels are mediated by maternal employment. We surmise that the linkage between country of origin and self-efficacy can hinge upon familial background. Specifically, we investigate whether self-efficacy gaps between individuals with employed mothers and those with homemaker mothers differ between the countries at scrutiny.*

## 4. Data and Methods

The sample consisted of 43 participants (28 men and 15 women) in an RE international program who responded to a survey designed for this research. We are conscious that the number of participants in our analysis did not allow us to provide implications on the national-level differences in self-efficacy in Argentina and Guatemala. However, this small-scale study aimed to provide an in-depth analysis of intersectional differences in self-efficacy of students that participated in international STEM programs, such as the DIEGO Erasmus+ Program.

Data were collected using an online survey between April and May 2019 distributed to the students of DIEGO via e-mail. They were kindly asked to respond to the survey, and 43 out of 125 participants responded to the questionnaire. We constructed a database with the information provided in the survey, and computed self-efficacy indices. The econometric analysis of this database was conducted using parametric tests, such as the *t*-test and one-way ANOVA. Some sensitivity checks included other choices of estimators to study the robustness of the main results.

The average age of the participants was 31 years; the average age for men (33 years) was slightly higher than that for women (27 years). All the participants had been awarded a bachelor's or a higher degree.

Our key outcome variable was self-efficacy, measured by means of the New General Self-Efficacy scale designed by Chen et al. (2004), which we further detail in what follows. We focused on three main groups of participants. We divided our sample first between women and men, second by country of origin, and third by maternal employment. Table 2 provides a summary of self-efficacy levels in the sample of the three main groups, along with income status and educational attainment.

**Table 2.** Self-efficacy mean levels by group.

| Variable | Subgroup | *No.* | Mean | SD | Min | Max |
|---|---|---|---|---|---|---|
| Whole sample | | 43 | 4.36 | 0.52 | 3 | 5 |
| Gender | Women | 15 | 4.42 | 0.53 | 3 | 5 |
| | Men | 28 | 4.32 | 0.51 | 3.13 | 5 |
| Education level | BA/BSc * | 25 | 4.31 | 0.56 | 3 | 5 |
| | Master | 11 | 4.48 | 0.52 | 3.25 | 5 |
| | Other | 6 | 4.3 | 0.43 | 3.63 | 4.86 |
| | Ph.D. | 1 | 4.69 | 0.44 | 4.38 | 5 |
| Country | Argentina | 23 | 4.13 | 0.51 | 3 | 4.86 |
| | Guatemala | 20 | 4.59 | 0.44 | 3.375 | 5 |
| Maternal employment | Employed | 32 | 4.32 | 0.47 | 3.13 | 5 |
| | Homemaker | 11 | 4.46 | 0.63 | 3 | 5 |
| Income status | Lower than average | 4 | 4.1 | 0.52 | 3.25 | 4.63 |
| | Average | 32 | 4.41 | 0.56 | 3 | 5 |
| | Higher than average | 3 | 4.25 | 0.14 | 4.125 | 4.38 |
| | No answer | 4 | 4.38 | 0.41 | 3.86 | 5 |

* Bachelor of Arts (BA), Bachelor of Science (BSc).

## 4.1. Measurement of Self-Efficacy

Self-efficacy can be understood as being task-specific or domain-specific (Luszczynska et al. 2005). Additionally, new research has provided a variety of scales to measure self-efficacy in other domains, such as occupational self-efficacy, as in Rigotti et al. (2008). In this paper, we focused on a generalized sense of self-efficacy, which refers to an individual's global confidence in their coping ability across a variety of scenarios. We opted for a general self-efficacy, since the students in our sample came from different fields of study, such as engineering, agriculture, or education. Thus, we considered that measuring self-efficacy in engineering or in RE would bias our analysis, since those who possessed a bachelor's degree in that precise field of study might be more likely to show higher levels of task-specific self-efficacy.

Our survey included the New General Self-Efficacy scale (NGSE), which was scored using a 5-point Likert-type scale, from strongly disagree (1) to strongly agree (5), on 8 questions (Chen et al. 2001, 2004). Self-efficacy tests such as the NGSE are used to measure if a certain teaching method could have an effect on the self-efficacy of adult learners (Croasmun and Ostrom 2011). Table 3 shows the mean and median of each item. The NGSE was computed as the arithmetic mean of these 8 items.

**Table 3.** Items of the New General Self-Efficacy scale.

| New General Self-Efficacy Scale | Mean | SD |
|---|---|---|
| I will be able to achieve most of the goals that I set for myself | 4.41 | 0.77 |
| When facing difficult tasks, I am certain that I will accomplish them | 4.4 | 0.68 |
| In general, I think that I can obtain outcomes that are important to me | 4.58 | 0.61 |
| I believe I can succeed at most any endeavor to which I set my mind | 4.33 | 0.83 |
| I will be able to successfully overcome many challenges | 4.52 | 0.68 |
| I am confident that I can perform effectively on many different tasks | 4.46 | 0.71 |
| Compared to other people, I can do most tasks very well | 3.93 | 0.78 |
| Even when things are tough, I can perform quite well | 4.21 | 0.77 |

## 4.2. Analysis

We followed the standard literature in using parametric tests, such as the *t*-test to compare mean differences in self-efficacy when considering two groups (gender, country, or maternal employment), and one-way ANOVA tests to check the consistency of the

results. To consider the intersectionality between country (Argentina versus Guatemala) and maternal employment, we made use of two-way ANOVA tests. We ran the Shapiro–Wilk test and Bartlett's test to validate the model used (see Shapiro and Wilk 1965; Arsham and Lovric 2011 for further methodological details).

## 5. Results

We tested all three hypotheses described in Section 3, confirming all our assumptions with the empirical analysis. In detail:

**Proof of Hypothesis 1.** We confirmed that there were no gender differences in self-efficacy for post-graduate students in RE. □

Women in our sample showed slightly higher self-efficacy than men. We first looked at potential gender differences using the whole sample ($n$ = 43), which was divided among 28 men (mean = 4.32, SD = 0.51) and 15 women (mean = 4.41, SD = 0.53). We conducted an independent $t$-test to study gender differences in mean self-efficacy, and we found no statistical significance ($t$ (43) = −0.6138, $p$ = 0.5423). To corroborate this lack of gender difference, we also performed a one-way ANOVA test, which again yielded no significant self-efficacy differences between women and men in our sample of students (F (1,46) = 0.38, $p$-value = 0.5423).

**Proof of Hypothesis 2.** We found significant differences between countries in the self-efficacy of students. □

We performed a $t$-test to analyze whether there were significant cross-country differences in self-efficacy. Considering the data on self-efficacy by country, the results suggested that this difference was statistically significant, and Guatemalans showed higher levels of self-efficacy ($t$ (41) = 3.147, $p$-value = 0.003).

Given the background differences in higher education and economic and social indicators between Argentina and Guatemala, this evidence could be the result of more difficult access to higher education in Guatemala (the gross enrolment ratio in tertiary education was just 22%) than in Argentina.

**Proof of Hypothesis 3.** We found weak evidence of higher levels of self-efficacy for students with employed mothers. □

We conducted an independent $t$-test using the whole sample to explore differences in mean group self-efficacy on the basis of maternal employment. Our measure of maternal employment was a dummy variable that took the value of 1 when an individual's mother currently was or had been in the labor market, and zero otherwise (respondents were asked to choose the option that best fit their mothers' labor market status, and were provided the following categories: (1) Paid worker; (2) Self-employed; (3) Unemployed; (4) Retired; (5) Housewife; (6) Other; to construct our dummy variable of maternal employment. We grouped categories 1 to 4 together under the type of "employed mother", whereas "housewife" was considered as homemaker mother; those individuals who responded "other" were excluded from the analysis. We obtained some weak evidence that maternal employment groups differed in their means of self-efficacy, with a level of significance lower than 5% (t(43) = 2.2453, $p$-value = 0.0298). In any case, it showed that individuals with homemaker mothers had higher self-efficacy than those with employed mothers. The results also were found using a one-way ANOVA test (F (1,44) = 5.04, $p$-value = 0.0298).

**Proof of Hypothesis 4.** We found that maternal employment played a mediating role in country differences in students' self-efficacy. □

We conducted a two-way ANOVA analysis to inspect differences in mean levels of self-efficacy between maternal employment in the two countries under scrutiny. This test is used when comparing more than two variables, which in our case were maternal employment (individuals with employed or homemaker mothers) and country of origin (Argentina versus Guatemala; see Table 4 for dimension of the sample analyzed).

**Table 4.** Country and maternal employment.

| Country | Maternal Employment | *No.* of Obs. | Mean | SD |
|---------|--------------------|---------------|------|-----|
| Argentina | Employed | 19 | 4.22 | 0.46 |
| Argentina | Homemaker | 4 | 3.69 | 0.56 |
| Guatemala | Employed | 12 | 4.44 | 0.49 |
| Guatemala | Homemaker | 8 | 4.81 | 0.24 |

The result associated with the independent variable of maternal employment (Table 5) was not statistically significant, and therefore, as in our previous step, we did not find meaningful differences based solely on family background. Interestingly, the interaction between country and mother's labor market status was statistically significant; thus, the differences in country-mean levels of self-efficacy hinged upon the mother's labor market status ($F = 7.77$, $p$-value = 0.0008). Therefore, our findings suggested that maternal employment might play a mediating role in country differences in self-efficacy in the sample.

**Table 5.** Results of the two-way ANOVA test.

| Variable | Partial SS * | df | F | Prob > F |
|----------|-------------|-----|-----|----------|
| Country | 1.96 | 1 | 18.89 | 0.0001 |
| Maternal employment | 0.034 | 1 | 0.32 | 0.5723 |
| Interaction | 0.803 | 1 | 7.77 | 0.0085 |
| Model | 2.19 | 3 | 7.04 | 0.0008 |
| Number of obs. = 39; R-squared = 0.3764; MSE ** = 0.322087; Adj R-squared = 0.3230. | | | | |

* Partial Sum-of-squares; ** Mean squares.

Figure 1 presents these estimates, which suggested that the interplay between country and maternal employment in self-efficacy did not work in the same way in Guatemala and Argentina. Although we did not want to extend the results using a two-way ANOVA test due to the limitations of our data, this tentative step showed how maternal employment interacted differently with country of origin in its impact on self-efficacy. Guatemalan students in the sample possessed higher self-efficacy than the Argentinian students. Further, we might consider that Argentinian students with employed mothers possessed higher self-efficacy than Argentinian students with homemaker mothers. In the case of Guatemalan students, we saw the opposite: the mean self-efficacy of individuals with homemaker mothers was higher than that of individuals with employed mothers.

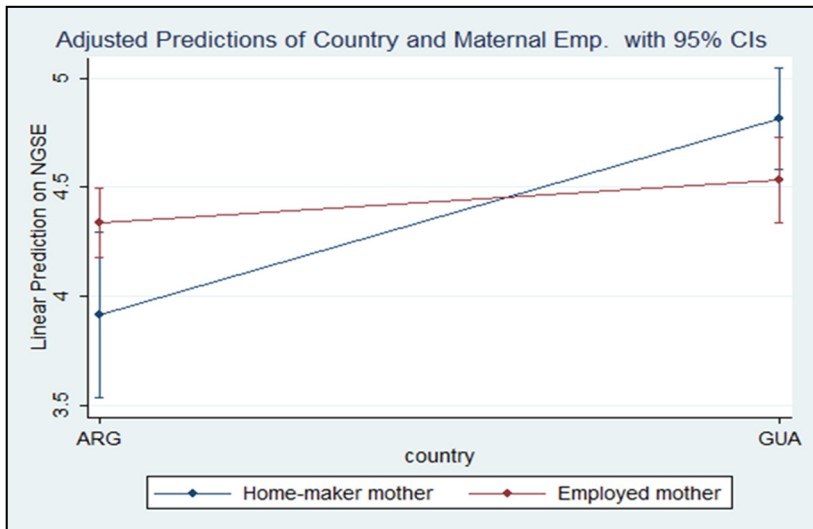

**Figure 1.** Self-efficacy differences in country and mother's employment status. Note: the figure refers to values at 95% confidence level (CI). In the ordinate axis the predicted value of the New General Self-Efficacy scale (NGSE). Referring to the countries ARG is Argentina and GUA is Guatemala.

## 5.1. Sensitivity Checks

We conducted two additional tests to provide additional support for the pattern we found. Our first sensitivity check considered whether the interaction between country of origin and employment status of the father also was significant. As in the previous step, we used a two-way ANOVA test to see whether country differences interplayed with the father's status in the labor market (Table 6). In this case, the model suggested significant differences in mean self-efficacy between the two countries, but these differences were not mediated through the parental role model derived from individuals' labor market participation. This result might suggest that the role of mothers in the construction of self-efficacy was more decisive than that of fathers.

**Table 6.** Fathers' employment estimates.

| Variable | Partial SS * | df | F | Prob > F |
|---|---|---|---|---|
| Country | 2.344 | 1 | 13.50 | 0.0008 |
| Father employment | 0.054 | 1 | 0.32 | 0.5775 |
| Interaction | 0.129 | 1 | 0.74 | 0.3941 |
| Model | 2.476 | 3 | 4.75 | 0.0067 |
| Number of obs = 41; R-squared= 0.2782; MSE ** = 0.416755; Adj R-squared = 0.2197. | | | | |

* Partial Sum-of-squares; ** Mean squares.

Our second sensitivity check and last estimation considered whether other maternal characteristics played a role in cross-country differences in the sample, such as educational attainment. Table 7 shows the four groups we considered, namely Argentinian students with mothers with a higher education level (we used the International Standard Classification of Education 2013 classification of educational attainment), Argentinian students whose mothers did not have a higher education degree, and the same for Guatemalan participants.

**Table 7.** Country and maternal education level.

| Country | Maternal Education Level | *n.* of Obs. | Mean | SD |
|---|---|---|---|---|
| Argentina | Higher education level (HE) | 7 | 4.38 | 0.3 |
| Argentina | Lower than HE | 16 | 4.02 | 0.55 |
| Guatemala | HE | 13 | 4.59 | 0.38 |
| Guatemala | Lower than HE | 7 | 4.59 | 0.58 |

We conducted a two-way ANOVA test that yielded similar results regarding country mean self-efficacy differences as aforementioned, and the outcome is displayed in Table 8: there was a statistically significant difference in self-efficacy between individuals in the sample from Argentina and Guatemala, by which the latter held higher scores for this precise construct. The educational attainment of the mother was not associated with a significant difference in self-efficacy, although the interaction between country and maternal education seemed to play a role in self-efficacy differences, at a significance level of 10%.

**Table 8.** Country and maternal education-level estimates.

| Variable | Partial SS * | df | F | Prob > F |
|---|---|---|---|---|
| Country | 1.867 | 1 | 11.47 | 0.0017 |
| Maternal education level | 0.016 | 1 | 0.10 | 0.7518 |
| Interaction | 0.546 | 1 | 3.35 | 0.0754 |
| Model | 2.881 | 3 | 5.90 | 0.0021 |
| Number of obs = 41; R-squared = 0.3236; MSE ** = 0.403446; Adj R-squared = 0.2687. | | | | |

* Partial Sum-of-squares; ** Mean squares.

Figure 2 presents these estimates and shows a similar picture as before. Among Argentinian students, an upgraded status of mothers was associated with higher self-efficacy, whereas among Guatemalan students, individuals with mothers with a lower education level showed higher self-efficacy than those whose mothers had a higher education degree. In sum, this provided further support for our previous results and for the argument that familial background, specifically the characteristics of mothers, matters to the social development of individuals along the entire academic and professional life cycle.

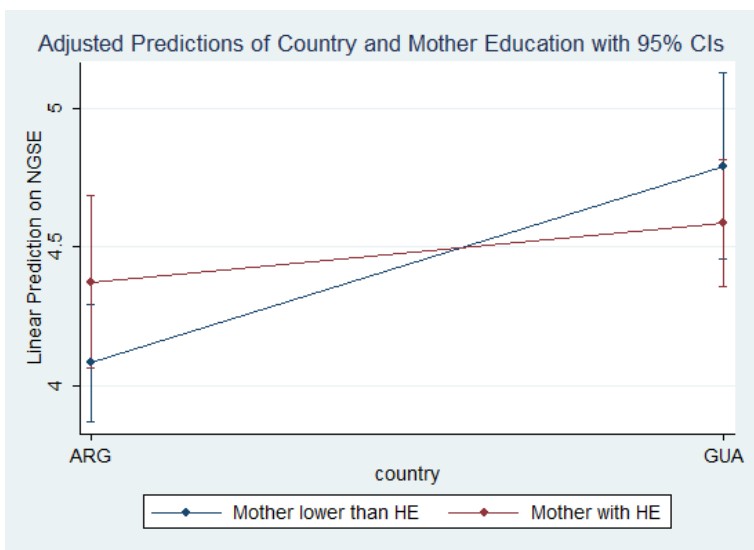

**Figure 2.** Self-efficacy differences in country and mother's higher educational level (HE). Note: the figure refers to values at 95% confidence level (CI). In the ordinate axis the predicted value of the New General Self-Efficacy scale (NGSE). Referring to the countries ARG is Argentina and GUA is Guatemala.

Before providing some tentative explanations for the results we obtained, it is worth highlighting the limitations of the analysis. This study was conducted using a sample of participants in an Erasmus+ training program co-funded by the European Union. Thus, the size of the sample was rather small, which forced us to strike a note of caution regarding the external validity of the results provided. However, participants in our sample were either working employees or students, and mainly at mid-career. This might contribute to existing literature, as self-perception studies commonly use college students in the early stages of their educational paths, as in Whiston et al. (2017). Additionally, we believe that our region of interest, Latin America, has been somehow unexplored in existing literature in comparison with Western countries (Roy et al. 2019). Finally, due to the cross-sectional nature of the information employed here, we discarded any causal interpretation of the patterns and differences in self-efficacy that we showed. Nevertheless, the study did not focus on the causal interpretation of drivers of self-efficacy but, rather, it was a descriptive analysis and comparison between different groups. As stated in Loeb et al. (2017), descriptive analysis often provides prima facie evidence necessary for subsequent argumentation of mechanism or causal relations.

Our sample of students did not show gender differences in self-efficacy. The absence of such differences might indeed be an important result, as it could indicate that women in male-dominated fields, as was the case with engineering of RE, regarded themselves as "exceptional women", so their self-perception of their own cognitive ability was similar to those of their male colleagues. This argument of "pioneer" women was elaborated in Charles and Bradley (2009), also showing how the phenomenon could be even stronger in countries where female representation in the overall higher education system was low. Indeed, this equality in self-efficacy levels of women and men in the sample was consistent with previous research, as in Pajares (2002) and Huang (2013).

Our estimates suggested significant country differences, whereby Guatemalan participants held higher levels of self-efficacy than did Argentinian participants. Following the findings in Morony et al. (2013), we explored potential mechanisms at work behind these differentials, considering family backgrounds (i.e., educational level of father or mother). We, in fact, provided some empirical support for the mediating role in self-efficacy that maternal employment might play. Descriptively, we saw how, among Guatemalans, students with homemaker mothers held higher self-efficacy than those with employed mothers. In contrast, Argentinian students in the sample whose mothers currently were or had been employed in the paid labor market held higher self-efficacy than those whose mothers did not work.

It is worth noting that the Guatemalan female labor force was lower than that of Argentina, whereas its level of gender inequality was higher. In that respect, Lundberg and Frankenhaeuser (1999) suggested that dual-earner families were more successful and beneficial for children's self-efficacy, provided fathers assumed an active, equal share of the household duties. Hence, in more gender-unequal countries, the gender mandates for mothers might penalize employed mothers or/and overburden women with paid and unpaid chores, resulting ultimately in self-efficacy losses in their children. It might be that Argentina, where the level of gender equality, female labor-force participation, and education attainment were relatively higher, evolved a transition toward more gender-egalitarian social values, which eventually allowed maternal employment to foster self-efficacy levels. It could also be the case that embedded gender social norms at the country level played a crucial role in shaping younger self-efficacy levels, creating a lower level of confidence in individuals whose mothers did not conform to the social norms.

Finally, the sensitivity checks conducted above suggested that the role that mothers played in the formation of self-efficacy of children might have long-lasting effects. Recall that the students in our sample were early- to mid-career individuals. Hence, identifying self-efficacy differences based on familial background might indeed show the magnitude of the impact of mothers—in terms of both labor participation and educational attainment—on the socialization process of individuals. Second, the sensitivity analysis provided further

insight regarding how the characteristics of mothers, and not those of fathers, were crucial in the development of social cognitive constructs.

*5.2. Limitations*

A note of caution should be struck in interpreting the results above. First, due to the limited number of participants in the survey, the results cannot be generalized to other contexts or at national level. Since this study was carried out on students from Argentina and Guatemala participating in a renewable energy international program, it would therefore be helpful to gather comparative data in other international programs, both in terms of country of origin and field of study. Second, the issue of self-selection in both participating in the program and in the survey should be taken into consideration. In this way, the answers may have been biased by high motivation of the respondents.

Despite the above-mentioned limitations, this study offered a thorough exploration of these students' self-efficacy levels and the potential socio-demographic factors influencing them. Future research of a longitudinal nature would complement the present paper by considering the change in self-efficacy in students after completion of training programs.

## 6. Conclusions

The paper focused on differences in self-efficacy among participants in an international higher education program on renewable energy, to advance our understanding of how self-efficacy depended on students' personal characteristics and social conditions.

We performed tests of equality of distribution, medians, and means of self-efficacy using information from a sample of students, considering their gender, country of origin, and parents' employment status. The results of our analysis were interesting when also considering the positive role of self-efficacy among students in terms of higher economic returns of education and labor-market outcomes. Oddly, in our sample we did not find differences in self-efficacy levels between women and men, and this was thanks to both the international environment and the innovative field of study, renewable energy, that seemed to mitigate the marginalization of women in STEM (science, technology, engineering, and mathematics) studies, serving as an amplifier for their empowerment and self-confidence. In fact, international programs in higher education, such as Erasmus+ programs and other international mobility programs for students, are increasingly attracting the attention of educational researchers and social scientists (Klemenčič et al. 2017; Iriondo 2019).

However, we found that social environment and gender norms, expressed mainly in terms of female labor-force participation and education attainment, had an impact on self-efficacy of younger individuals, since they shape social categorization and the structure of power within society. The findings of our empirical analysis in fact suggested cross-country differences in self-efficacy, in particular when we explored the intersection among students' countries of origin, family backgrounds, and the incidence of gender norms in specific social contexts. Moreover, we found a crucial role of the characteristics of mothers, in particular when looking at their employment status in formal labor markets, in the development of social cognitive constructs of younger generations.

This inter-generational transmission of self-efficacy through maternal role models is of paramount importance in understanding the complexity of the mechanisms to also change existing social inequalities in self-perception in different countries and labor markets. As lines of future research, we propose replicating a similar analysis using a greater coverage of data, which might allow for better understanding of patterns and causal linkages among different generations and social contexts.

**Author Contributions:** Conceptualization, M.C., G.Z., and I.Z.; methodology, I.Z.; data curation, G.Z. and I.Z.; writing—original draft preparation, I.Z.; writing—review and editing, G.Z.; supervision, M.C. All authors have read and agreed to the published version of the manuscript.

**Funding:** This research was funded by Sapienza University of Rome under the IRENE (Impact of the gReen energy EducatioN on job creation, women Empowerment and internationalization) program, grant number RG11715C821B6C63.

**Institutional Review Board Statement:** Not applicable.

**Informed Consent Statement:** Informed consent was obtained from all subjects involved in the study.

**Acknowledgments:** The authors are deeply grateful to Gabriella Calderari and Katiuscia Cipri for their help in collecting information in the context of the DIEGO program; they are also grateful to Anna Conte (IRENE coordinator) for the financial support.

**Conflicts of Interest:** The authors declare no conflict of interest. The funder had no role in the design of the study; in the collection, analyses, or interpretation of data; in the writing of the manuscript; or in the decision to publish the results.

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
