# Peer review of "Intersectional Gaps in Self-Efficacy among Post-Graduate Students in International Renewable-Energy Programs: The Role of Maternal Employment"

_socsci, doi:10.3390/socsci10070242_

Round 1

Reviewer 1 Report

The number of study participants is far too small to make statements about the differences between the two countries in terms of self-efficacy. The work can be redone by considering a significant sample.

Reviewer 2 Report

Great job about the manuscript.

Please:

1- Change abstract section in order to journal sections

2- Add limitations of the study

3- Describe a bit more the procedure section

4- How the sample was collected?

5- If it was posible, would be convenient to reduce the hypotheses

Round 2

Reviewer 1 Report

The authors made the requested clarifications regarding the limitations of the research results.

Author Response

Thanks, we are happy that the new version of the manuscript correctly answered to your previous comments.

Best regards, 

the authors